# Novel Choline-Deficient and 0.1%-Methionine-Added High-Fat Diet Induces Burned-Out Metabolic-Dysfunction-Associated Steatohepatitis with Inflammation by Rapid Immune Cell Infiltration on Male Mice

**DOI:** 10.3390/nu16234151

**Published:** 2024-11-29

**Authors:** Takatoshi Sakaguchi, Yasuharu Nagahama, Nanako Hamada, Shailendra Kumar Singh, Hayato Mikami, Kazuhiko Maeda, Shizuo Akira

**Affiliations:** 1Laboratory of Host Defense, Immunology Frontier Research Center, Osaka University, Suita 565-0871, Japan; sakag745@ifrec.osaka-u.ac.jp (T.S.); y-nagahama@ifrec.osaka-u.ac.jp (Y.N.); nanako11@ifrec.osaka-u.ac.jp (N.H.); singh-sk@ifrec.osaka-u.ac.jp (S.K.S.); kazmaeda@biken.osaka-u.ac.jp (K.M.); 2Host Defense Laboratory, Immunology Unit, Osaka Research Center for Drug Discovery, Otsuka Pharmaceutical Co., Ltd., Minoh 562-0029, Japan; 3Oriental Yeast Co., Ltd., Itabashi 174-8505, Japan; mikami.hayato@nisshin.com; 4Department of Host Defense, Research Institute for Microbial Diseases (RIMD), Osaka University, Suita 565-0871, Japan; 5Center for Advanced Modalities and Drug Delivery System (CAMaD), Osaka University, Suita 565-0871, Japan

**Keywords:** burned-out MASH, CDAHFD, hepatic inflammation, MASLD, mitochondrial dysfunction

## Abstract

**Background:** Metabolic-dysfunction-associated steatotic liver disease (MASLD) is a progressive liver disorder that possesses metabolic dysfunction and shows steatohepatitis. Although the number of patients is globally increasing and many clinical studies have developed medicine for MASLD, most of the studies have failed due to low efficacy. One reason for this failure is the lack of appropriate animal disease models that reflect human MASLD to evaluate the potency of candidate drugs. **Methods:** We developed a novel choline-deficient and 0.11%-methionine-added high-fat diet (CDAHFD)-based (MASH) diet that can induce murine metabolic-dysfunction-associated steatohepatitis (MASH) without severe body weight loss. We performed kinetic analyses post-feeding and proposed an appropriate timing of MASH pathogenesis by quantitatively analyzing steatosis, inflammation, and fibrosis. **Results:** This MASH diet induced liver fibrosis earlier than the conventional CDAHFD model. In brief, lipid accumulation, inflammation, and fibrosis started after 1 week from feeding. Lipid accumulation increased until 8 weeks and declined thereafter; on the other hand, liver fibrosis showed continuous progression. Additionally, immune cells, especially myeloid cells, specifically accumulated and induced inflammation in the initiation stage of MASH. **Conclusions:** The novel MASH diet promotes the dynamics of lipid deposition and fibrosis in the liver, similar to human MASH pathophysiology. Furthermore, immune-cell-derived inflammation possibly contributes to the initiation of MASH pathogenesis. We propose this model can be the new pre-clinical MASH model to discover the drugs against human MASH by evaluating the interaction between parenchymal and non-parenchymal cells.

## 1. Introduction

Metabolic-dysfunction-associated steatotic liver disease (MASLD) and metabolic dysfunction associated steatohepatitis (MASH) are the current terms for liver disease related to metabolic dysfunction and steatosis that have been previously described as non-alcoholic fatty liver diseases (NAFLDs) and non-alcoholic steatohepatitis (NASH), respectively [1]. The current epidemiology estimates that 30% of the global population has MASLD, and approximately 20% of MASLD patients have MASH [1]. A clinical problem in MASH is that it may develop into cirrhosis and MASH-related hepatocellular carcinoma [2]. Although many clinical studies have been investigated to develop compounds to treat MASH, most have failed because of poor efficacy [3,4]. One reason for this clinical failure is the lack of pre-clinical animal models that mimic human MASH and predict its efficacy. To address this, many studies have been demonstrated to develop appropriate animal models for human MASH [5]. A recent study ranked these models based on their similarity to human MASH but concluded that none perfectly replicate it [6]. The Western diet (WD) better reflects the metabolic aspects of the disease. In contrast, the choline-deficient diet mimics the fibrosis-promoting component, highlighting the importance of choosing an appropriate diet based on the purpose.

The choline-deficient and 0.1%-methionine-added high-fat diet (CDAHFD) was developed as a MASH model to analyze the pathogenesis of human MASH [7]. CDAHFD has a milder effect on body weight changes but develops into fibrosis rapidly compared to the high-fat diet (HFD). The NAFLD activity score (NAS) and fibrosis stage are semi-quantitative scoring systems that evaluate the grades of steatosis (score 0–3), lobular inflammation (score 0–3), ballooning (score 0–2), and fibrosis (stage 0–4) in human MASLD and murine MASLD models [8,9]. The CDAHFD causes score 2 steatosis within 1 week based on oxidative-stress-induced mitochondrial dysfunction [10,11,12]. In this model, fibrosis is observed after 6 weeks of feeding [7]. Tumor formation can be seen after 36 weeks with nodular regenerative hyperplasia, adenomas, and hepatocellular carcinoma [13]. In addition, CDAHFD-fed humanized-liver chimeric mice show similar histological changes with human MASH patients [14]. These reports indicate that the CDAHFD is suitable for analyzing mitochondrial dysfunction and useful for investigating the early stage of human MASH. However, CDAHFD feeding inhibits weight gain, and this phenotype is apart from human MASH pathophysiology [6,7].

Immune cells influence the pathogenesis of MASH through inflammation and tissue repair [15]. Although neutrophils infiltrate the liver and induce inflammation in the early stages of MASH, they do not contribute to fibrogenesis [16,17,18,19,20]. In addition, B cell subsets have different properties and opposite contributions to the pathogenesis of MASH [21,22,23]. Many studies have focused on specific cells at particular stages, but since MASH is progressive, the role of immune cells may vary throughout its progression. This gap may lead to the misinterpretation of experimental results and drug efficacy predictions. Therefore, it is essential to trace the comprehensive immune cell transition in MASH pathogenesis and to identify the appropriate period for assessing the contribution of immune cells.

In this study, we developed a novel MASH diet with some changes from the ingredients of the previously reported CDAHFD. Briefly, we added linoleic acid to induce inflammation and reduce the fat–calorie ratio to avoid body weight reduction. We showed the impact of our MASH diet on steatosis induction, inflammation, fibrosis, and serum components over time to identify the optimal period to analyze MASH development. Lipid accumulation decreased after 24 weeks, like burned-out MASH in humans. Furthermore, we provide insights into the immune cell typing in the MASH diet by cytometry by time-of-flight (CyTOF) analysis and identify the rapid accumulation of myeloid cells in the liver. These results indicate the benefit of our MASH diet as a pre-clinical animal model to investigate the drugs for MASH treatment.

## 2. Materials and Methods

### 2.1. Animals and Experimental Protocol for the MASH Diet Model

Specific pathogen-free 5 weeks old male C57BL/6J mice were purchased from Japan SLC, Inc. (Shizuoka, Japan), and after 1 week of acclimation, mice were randomly divided into two groups (up to 3 mice per cage): the Chow group was fed a commercial standard diet (Chow; #CE-2, CLEA, Shizuoka, Japan); the MASH group was fed a modified choline-deficient and 0.11%-methionine-added high-fat diet (CDAHFD) (MASH; #OYC-NASH1, Oriental Yeast Co., Ltd., Tokyo, Japan). At 0, 1, 2, 4, 8, 12, 24, and 48 weeks after the feeding, the animals were weighed, euthanized by cervical dislocation, and the samples were collected for analysis (Figure 1). Blood was collected by cardiac puncture, clotted at room temperature for 30 min (#365978, BD Bioscience, Franklin Lakes, NJ, USA), centrifuged (6000× *g*, 4 °C, 90 s), and stored at −80 °C. For real-time PCR and Western blot analysis, small pieces of liver tissue were snap-frozen in liquid nitrogen and stored at −80 °C. For the histological analysis, the liver tissue was also fixed in 10% neutral buffered formalin (#060–01667, Fujifilm Wako, Osaka, Japan). All mice were housed under specific pathogen-free conditions (23 ± 2 °C, 55 ± 5% humidity, 12/12 h light/dark cycle) in the Animal Resource Center for Infectious Diseases at Osaka University. All animal experiments were performed with approval from the Animal Research Committee of the Research Institute for Microbial Disease, Osaka University.

#### Nutritional Composition of the MASH Diet

We modified the ingredients of the CDAHFD and generated the MASH diet. The ingredients of the MASH diet are summarized in Table 1.

### 2.2. Isolation of Immune Cells from Blood and the Liver

Red blood cells were removed by adding ACK lysing buffer (#A1049201, Thermo Fisher Scientific, Waltham, MA, USA) to the blood and incubating at RT for 2 min to isolate immune cells. Then, the cells were washed twice with CyFACS buffer (0.1% BSA + 2 mM EDTA + 0.1% Sodium Azide made with 1 × PBS) and resuspended in CyFACS buffer for CyTOF analysis. For immune cell isolation from the liver, the liver was finely chopped on ice and suspended in Buffer 1 (10% FBS + 25 mM HEPES made with RPMI-1640 with Glutamax (#61870-036, Thermo Fisher Scientific, pH 7.4). After centrifugation, the liver was suspended in Buffer 2 (10 μg/mL Liberase TM (#05401119001, Roche, Basel, Switzerland) + 2 U/mL DNase (#M0303L, New England Biolabs, Ipswich, MA, USA) + 3 mM CaCl_2_ + 25 mM HEPES made with RPMI-1640 with Glutamax), incubated at 37 °C with shaking (1 cycle/s) for 25 min and put on ice for 5 min. Then, the cell suspension passed through a 100 μm filter, was centrifuged, and resuspended in Buffer 3 (33% Percoll (#17089102, GE Healthcare, Chicago, IL, USA) + 1 U/mL Heparin (#A493, Mochida Pharmaceutical Co., Ltd., Tokyo, Japan) made with HBSS (#17460-15, Nacalai Tesque, Kyoto, Japan)). After 30 min of centrifugation, the supernatant was removed, and red blood cells were lysed with ACK buffer. Finally, the cells were resuspended in CyFACS buffer.

### 2.3. CyTOF Analysis

A single-cell suspension of 1–2 × 10^6^ cells was used for CyTOF analysis. Briefly, cells were washed twice with CyFACS buffer. After Fc-blocking for 30 min, the cell surface protein was stained with a heavy-metal-labeled antibody cocktail (#201306, Standard BioTools, South San Francisco, CA, USA) for 45 min at RT. Next, dead cell staining was performed with cell-ID Cisplatin-198Pt (#201198, Standard BioTools) for 5 min at RT. After washing twice, cells were fixed in Maxpar Fix and Perm Buffer (#201067, Standard BioTools) overnight at 4 °C. The next day, cells were incubated with intercalator-Ir (#201192, Standard BioTools) in Maxpar Fix and Perm Buffer at RT, washed twice with MilliQ (Merck Millipore, Darmstadt, Germany), and resuspended in Maxpar Cell Acquisition Solution (#201240, Standard BioTools) to 2.5–5 × 10^5^ cells/mL. Cells were analyzed using the Helios CyTOF system (Standard BioTools), and data were collected and analyzed by FlowJo and Cytobank following the manufacturer’s protocol.

### 2.4. Biochemical Analysis

Serum samples were shipped to Oriental Yeast Co., Ltd. (Tokyo, Japan), and biochemical analysis was performed using BioMajesty™ (#JCA-BM6050, JEOL Ltd., Tokyo, Japan).

### 2.5. Liver Histology

For hematoxylin and eosin (H&E) staining and Sirius red staining, the formalin-fixed liver was embedded in paraffin and sectioned at 5 μm. After deparaffinization and hydration, sections were stained with Hematoxylin (#MHS16, Sigma-Aldrich, St. Louis, MO, USA) for 3 min, washed with running water, and stained with Eosin (#051-06515, Fujifilm Wako) for 3 min. Sirius red staining was performed by a Picrosirius Red Stain Kit (#24901-250, Polysciences, Warrington, PA, USA) following the manufacturer’s protocol. For Oil Red O staining, 4% paraformaldehyde in PBS-fixed liver was transferred to 30% sucrose in PBS until the tissue sank at 4 °C, embedded in a Tissue-Tek O.C.T. compound (#4583, Sakura Finetek, Tokyo, Japan). The liver was sectioned at 10 μm, dried at RT, and sections were immersed in Oil Red O staining solution (0.3% Oil Red O (#25633-92, Nacalai Tesque) in isopropanol: MilliQ (3:2)) for 15 min. Images were acquired using a VS-200 slide scanner (Evident, Tokyo, Japan) and a BZ-X810 microscope (Keyence, Osaka, Japan). Quantitative measurements of the images were carried out with ImageJ software (ImageJ 1.54f, National Institute of Health, USA). At least 10 images were randomly taken from each section for all quantitative measurements and scoring. NAFLD Activity Score (NAS) was determined by veterinarians specialized in pathology and experts from SMC laboratories (Tokyo, Japan) and Oriental Yeast following previous reports [9,10].

### 2.6. Real-Time PCR

Total RNA was extracted from the liver tissue using TRIzol (#15-596-018, Thermo Fisher Scientific) following the manufacturer’s protocol. RNA was treated with DNase (#M0303L, New England Biolabs) and purified using a Monarch RNA cleanup kit (#T2030, New England Biolabs). Reverse transcription was performed using ReverTra Ace qPCR RT Master Mix (#FSQ-201, TOYOBO, Osaka, Japan). THUNDERBIRD SYBR qPCR Mix (#QPS-201, TOYOBO) was utilized to conduct real-time PCR on QuantStudio 6 Pro (Applied Biosystems, Waltham, MA, USA). *Hprt* was used as an internal control. The list of probes is shown in Table 2.

### 2.7. Western Blotting

The liver was lysed in RiPA buffer (50 mM HEPES + 150 mM NaCl + 0.125 μM EDTA + 0.1% SDS + 0.5% Sodium Deoxycholate + 0.5% NP-40 + 0.1% Tween20 made with MilliQ). The protein amount in the lysate was measured by a Pierce BCA Protein Assay Kit (#23225, Thermo Fisher Scientific) according to the manufacturer’s protocol. After loading 10 μg of protein for SDS-PAGE, the protein was transferred to the PVDF membrane, followed by Western blotting. Target proteins were detected by the antibodies as listed below: COL1A1 (#ABT123) and SREBP-1 (#MABS1987) from Sigma-Aldrich; and α-SMA (#ab5694), PGC-1α (#ab191838), Parkin (#ab77924), and LC3B (#ab192890) from Abcam (Cambridge, UK); and PGC-1β (#22378-1-AP) from Proteintech (Rosemont, IL, USA); and AceCS1 (#3658), ACL (#4332), p-ACL (#4331), ACC (#3676), p-ACC (#11818), Fas (#3180), AMPK (#5831), p-AMPK (#2535), PPARγ (#2435), and Tom20 (#42406) from Cell Signaling Technology (Danvers, MA, USA); and β-actin (#sc-47778) from Santa Cruz (TX, USA).

### 2.8. Statistical Analysis

Statistical analysis of the experimental data was performed in GraphPad Prism version 10 using appropriate tests, as indicated in figure legends, with significant differences in markers on all figures.

## 3. Results

### 3.1. MASH Diet Feeding Induced Hepatic Steatosis, Inflammation, and Tumorigenesis

Six-week-old male mice were randomly subjected to either the Chow or the MASH diet for up to 48 weeks (Figure 2A). In the MASH diet, body weight loss was limited to less than 5% until 12 weeks, while 20–25% of the body weight loss started after 12 weeks, similar to the conventional CDAHD (Figure 2B). In addition, tumor formation was observed in 60% of the mice at 48 weeks (Figure 3C,D). These tumors were hepatocellular adenomas or nodular regenerative hyperplasia with macrovascular steatosis, not hepatocellular carcinoma (Figure 2E).

While score 1 steatosis occurs just after 1 week in the CDAHFD [10], the MASH diet feeding induced score 2 steatosis within 1 week and showed score 3 after 2 weeks (Figure 2F,G). Similar to the CDAHFD, the MASH model showed no ballooning of hepatocytes. Lobular inflammation showed score 2 at 2 weeks and sustained until 48 weeks (Figure 2F,G). NAS showed score 2 at 1 week and reached score 5 after 2 weeks (Figure 2G). AST and ALT, markers for liver injuries, were increased from 1 week, and thereafter were highly produced during the entire period (Figure 2H).

### 3.2. MASH Diet Feeding Induced Hepatic Fibrosis in a Time-Dependent Manner

Liver fibrosis was observed after 4 weeks, and collagen-deposited region spread after 8 weeks of MASH feeding (Figure 3A,B). Fibrosis scores ranged from 1 to 2, and no bridging formation was observed at week 48 (Figure 3A). In the liver, collagen is produced by activated α-SMA-positive hepatic stellate cells [24]. The expression of COL1A1 increased from 4 weeks after MASH feeding. Though α-SMA also increased from 4 to 8 weeks, its expression decreased after 12 weeks (Figure 3C,D). Additionally, *Col1a1*, *Acta2*, and *Tgfb1* expression increased after 1 or 2 weeks of MASH induction (Figure 3E). These results suggest that hepatic fibrosis develops rapidly within 4 weeks and is associated with hepatic stellate cell activation.

### 3.3. MASH Diet Feeding Induced Inflammation in the Early Stage

Since hepatic stellate cell activation contributes to liver fibrogenesis [25], we next performed immuno-phenotyping of CD45^+^ cells by using CyTOF. CD45^+^ immune cell accumulation started increasing 1 week after MASH feeding and gradually diminished (Figure 4A). Interestingly, the number of myeloid cells, such as granulocytes and macrophages, increased in the liver, hence CD4^+^ T cells were decreased at 1 week (Figure 4B,C). At week 4, the proportion and actual number of granulocytes and macrophages diminished in the liver compared to that at 1 week (Figure 4B,C). The exact number of most of the immune cells decreased in the liver except for macrophages at 8 weeks (Figure 4B,C). The proportion of granulocytes also increased in the blood at 1 week and decreased at 4 weeks and 8 weeks, similar to the dynamics in the liver (Figure 4B,C). In contrast, at 8 weeks, the proportion of B cells in the blood increased despite a decrease in the B cell population in the liver (Figure 4B,C). We then evaluated the mRNA expression of pro-inflammatory cytokines in the MASH liver. Interestingly, Tnf, Il1b, and Ifng expression increased rapidly, while Il6 was unchanged (Figure 4D). These results suggest that inflammation in the liver occurs after 1 week or earlier depending on the myeloid cell infiltration.

### 3.4. Effects of MASH Diet Feeding on Lipid Accumulation and De Novo Lipogenesis in the Liver

Since score 2 steatosis was observed 1 week after MASH diet feeding, we performed the kinetic analysis of lipid accumulation in the MASH liver by Oil Red O staining (Figure 5A). As expected, more than 10% lipid accumulation was observed after 1 week and 30% at 8 weeks (Figure 5B). However, lipid accumulation decreased after 24 weeks, indicating the development of burned-out MASH (Figure 5B). Although reduced secretion of VLDL from the liver has been proposed to cause lipid accumulation in the chorine-deficient diet model of MASH, how CDAHFD feeding affects de novo lipogenesis in the liver is unclear [26]. Therefore, we assessed the expression of enzymes involved in the de novo fatty acid synthesis. Acetate and citrate are converted to acetyl CoA by AceCS1 and ACL in the liver, respectively [27,28]. After the MASH feeding, the expression of these enzymes decreased from 1 week (Figure 5C,D). In addition, the expression of SREBP-1, the transcription factor for lipid synthesis, was also reduced (Figure 5C,E). Furthermore, the expression of AMPK, which suppresses lipid synthesis via the downregulation of SREBP-1, was increased and phosphorylated, indicating that AMPK was activated (Figure 5C,D,F) [29]. These results suggest that fatty acid synthesis was suppressed in the MASH diet by the activation of the AMPK signaling pathway.

### 3.5. Effects of MASH Diet on Serum Components

We performed a serum biochemical analysis to determine the effects of MASH feeding on serum components. There were no obvious differences in total protein, albumin, A/G, and iron (Table 3). Triglyceride decreased at 1 week and 48 weeks, but no difference was observed in other periods (Table 3). On the other hand, for cholesterol, E/T was reduced by MASH feeding throughout the entire period due to a decrease in cholesteryl esters or an increase in free cholesterol (Table 3). Moreover, LDL cholesterol was lower for 12 weeks and HDL cholesterol for 24 weeks from the beginning of feeding (Table 3). These results and the decreased expression of SREBP-1 in the liver suggested the accumulation of cholesterol in the liver.

### 3.6. Effects of MASH Diet on PGC-1α/β and Mitophagy

PGC-1α and PGC-1β are the master regulators of mitochondrial homeostasis [30,31]. Depending on the nutritional status, PGC-1α regulates glycogenesis and fatty acid oxidation, while PGC-1β regulates lipid synthesis, VLDL synthesis, and secretion. Additionally, both have equal regulatory capacity for mitochondrial biosynthesis [32]. Since the CDAHFD model elicits mitochondrial dysfunction and oxidative stress, we next investigated the effects of the MASH diet on mitochondria [10,11]. The expression of PGC-1α and PPARγ decreased during all periods except for 12 weeks (Figure 6A,B). Tom20, expressed on the mitochondrial outer membrane, decreased at all time points except for 4 and 12 weeks (Figure 6A,B). Meanwhile, Parkin, specifically involved in the mitophagy that degrades mitochondria, showed an increase only at week 12 (Figure 6A,B). LC3BII, an indicator of autophagosome formation, increased from 4 weeks after feeding on the MASH diet and decreased at 12 weeks. These results suggest that mitochondrial mass was reduced during the progression of MASH in correlation with PGC-1α expression in the liver. In addition, this mitochondrial mass reduction may be rescued by LC3BII-mediated autophagy. The decrease of PGC-1β was consistent with the lipid synthesis reduction during MASH diet feeding (Figure 6A,B).

Since mitochondrial mass reduction may affect the bioenergetic balance, we investigated the gene expression that functions in the metabolic pathway. *Pck2*, a marker of glycogenesis, and *Pdk4*, which is involved in fatty acid oxidation, increased in expression after MASH feeding (Figure 6C). Whereas the expression of *Cox4i1*, involved in OXPHOS, decreased during the entire period (Figure 6C), *Cox4i2* was compensatorily upregulated (Figure 6C), consistent with the previous report [33]. The expression of *Plin2*, involved in fat droplet formation, was unchanged (Figure 6C). These results suggest that MASH diet feeding reduces mitochondrial OXPHOS and biogenesis, but instead promotes fatty acid degradation and gluconeogenesis against fat burden.

## 4. Discussion

We investigated the novel CDAHFD-based MASH diet to induce liver steatosis and fibrosis in mice similar to the human MASH phenotype. In addition, we identified the possible contribution of immune cells to the MASH pathogenesis. This MASH diet can induce score 2 steatosis at 1 week and fibrosis with collagen deposition at 4 weeks after feeding. We determined to clarify the optimal feeding period for the analysis of steatosis, inflammation, and fibrosis to demonstrate its usefulness as a MASH disease model while indicating its advantages and disadvantages.

The CDAHFD induced about 20% body weight loss after 14 weeks, and continuous feeding up to 48 weeks resulted in about 25% body weight reduction compared to that of the Chow diet [7,13]. On the other hand, the body weight of the mice fed the MASH diet was almost the same as that of the Chow diet for up to 12 weeks. Although the MASH diet showed a similar pathophysiological process as the CDAHFD, the NAS reached five at 2 weeks, and the liver injury index increased markedly from 1 week, indicating that the MASH diet rapidly induced MASH more than the CDAHFD. Histological analysis showed that fibrosis in the liver increased from 4 weeks after MASH diet feeding and significantly increased at 8 weeks. On the other hand, the mRNA expression of *Col1a1* and *Acta2* increased from 1 week, indicating it may also be meaningful to analyze the onset of fibrogenesis from 1 week. In the early stage of human MASH, the expression of α-SMA in the liver positively correlates with that in the fibrosis stage [34,35,36,37]. Despite hepatic stellate cell activation impacting liver fibrosis, α-SMA expression decreased after 8 weeks in our MASH model, whereas fibrosis continued progression. Additionally, lobular inflammation was suppressed after 6 weeks in the CDAHFD, whereas it persisted for a long period (up to 48 weeks) in the MASH diet. These results indicate the pathophysiological difference between the CDAHFD diet and the MASH diet, and the immune cells that cause inflammation possibly promote fibrosis in the MASH diet.

Since the contribution of immune cells to MASH pathogenesis in the CDAHFD was unclear, we comprehensively analyzed the immune cells from the blood and liver to gain insights into the inflammatory response during the progression of MASH in the MASH diet. Briefly, hepatic immune cell infiltration was obvious at 1 week after feeding, with granulocytes, monocytes, and macrophages accounting for the greatest proportion. Importantly, the infiltration of neutrophils and monocytes has been identified in the early stages of human MASH [25]. These increased granulocytes, containing neutrophils, may contribute to the resolution of hepatic fibrosis [20]. In mice, B cells have been suggested to have both detrimental and protective effects in MASLD, whereas CD8^+^ T cells promote liver fibrosis resolution [23,38,39,40]. Since the contribution of B cells and T cells to hepatic fibrosis is still controversial, further detailed phenotyping is needed to determine their role in the MASH pathogenesis at several periods [21,22,23,25]. In the pathophysiology of MASH, TNF-α plays an important role in pathogenesis, while IL-1β causes inflammatory cell infiltration into the liver via inflammasome activation [37,41]. On the other hand, although IL-6 causes systemic insulin resistance in MASH, the *Il6* mRNA expression is unchanged in the liver of human MAFLD [42,43]. Although our study also demonstrated that *Il6* expression was unchanged in the liver during MASH pathogenesis, the contribution of IL-6 to the systemic insulin resistance associated with MASH should be further investigated.

Despite lipid synthesis being enhanced in human MASLD, some patients show a decrease in lipid droplets in advanced MASH, characterized as “burned-out MASH” [43,44,45]. As a choline-deficient diet inhibits lipid synthesis and VLDL secretion from the liver, the expression of enzymes involved in de novo lipogenesis and lipoprotein amounts in the serum decreased in our model [26]. In addition, since VLDL contains high levels of cholesterol, which is critical to establishing liver fibrosis, the inhibition of VLDL secretion induces VLDL accumulation in the liver and promotes fibrosis [46]. Furthermore, SREBP-1 expression decreases in burned-out MASH, and the inhibition of SREBP-1 increases the possible carcinogenesis in the liver via reduced autophagic flux [43,44]. In our model, SREBP-1 expression decreased after 1 week of feeding, and a reduction in lipid accumulation was observed from 8 weeks, indicating that the disease state of burned-out MASH might have started during this period.

Mitochondrial dysfunction leads to increased ROS production, which plays an important role in MASH pathogenesis [12]. In the CDAHFD model, mitochondrial function is impaired only 1 week after feeding [10]. In a healthy liver, damaged mitochondria are degraded by mitochondria-specific autophagy, known as mitophagy [47]. However, a MASH liver shows a disruption in mitophagy, causing the accumulation of damaged mitochondria and leading to MASH progression [48]. Since a decreased expression of SREBP-1 was observed in our MASH model, we hypothesized autophagic flux may affect MASH progression and investigated both mitochondrial degradation and biogenesis. In burned-out MASH, the decreased expression of LC3BII might indicate the suppression of macroautophagy. In addition, the constitutive reduction of *Cox4i1* expression during the MASH diet feeding periods implied the inhibition of mitochondrial respiration. These results are consistent with the previous reports on human advanced MASH [43,44]. Whereas the expression of PGC-1α is decreased in mild human MASH, macroautophagic flux is increased compared to simple steatosis [48]. Furthermore, PGC-1β promotes lipogenesis independently of PGC-1α in hepatocytes [49]. These data are consistent with results at 4 weeks in our MASH model. We further showed that PGC-1β expression decreased after 8 weeks, suggesting a transition to burned-out MASH with a decrease in accumulated lipids. In contrast, PGC-1α expression was increased in advanced MASH. These coordinated changes between PGC-1α and PGC-1β may influence the pathogenesis of burned-out MASH. However, further investigation is required to fully understand the hepatic lipid deposition status in MASH by PGC-1 balance.

## 5. Conclusions

In this study, we developed a modified MASH diet with inflammation, fibrosis, and the dynamics of lipid deposition in the liver, similar to the disease state of human MASH patients in some ways. Briefly, this MASH diet can induce fibrosis earlier than the conventional CDAHFD model with myeloid cell infiltration and inflammation in the liver. This new pre-clinical MASH model allows us to investigate the pathophysiology of MASH by evaluating the interaction between parenchymal and non-parenchymal cells.

## Figures and Tables

**Figure 1 nutrients-16-04151-f001:**
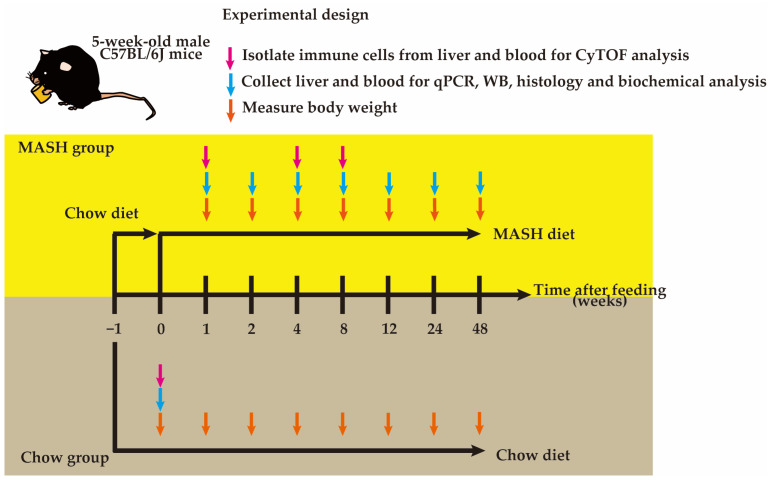
Schematic diagram of experimental design.

**Figure 2 nutrients-16-04151-f002:**
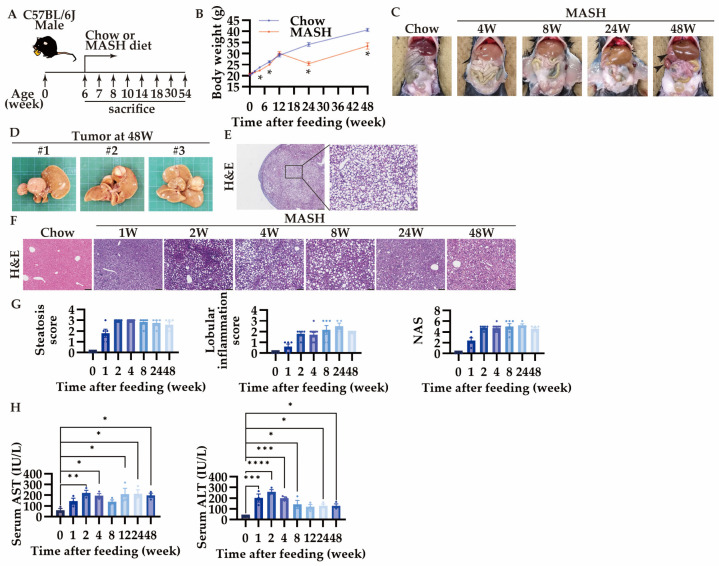
MASH diet feeding caused hepatic steatosis, inflammation, and tumorigenesis. (**A**) Schematic diagram of experimental procedure; (**B**) body weight (n = 5–6); (**C**) changes in the appearance of the digestive tract over time; (**D**) the appearance of the tumor; hematoxylin and eosin (**H**,**E**) staining of (**E**) the tumor (scale bars, 500 μm (left) or 100 μm (right)), and (**F**) the liver, scale bars, 100 μm; (**G**) scores of steatosis and lobular inflammation, and NAFLD activity score (NAS) (n = 4–10); (**H**) serum AST and ALT levels (n = 3). Zero weeks (0W) in the figure indicates Chow-diet-fed control mice. Values are presented as mean ± S.E.M. * *p* < 0.05, ** *p* < 0.01, *** *p* < 0.001, and **** *p* < 0.0001 vs. 0-week group. Statistical significance was calculated by a one-way ANOVA with Dunnett’s post hoc test.

**Figure 3 nutrients-16-04151-f003:**
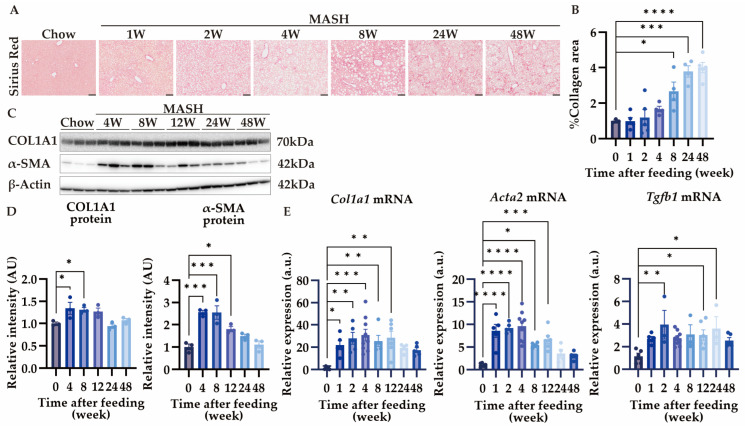
MASH diet feeding caused hepatic fibrosis. (**A**,**B**) Sirius red (SR) staining of the liver (n = 4–5); (**C**,**D**) the protein expression of COL1A1 and α-SMA in the liver (n = 3); (**E**) the mRNA expression of *Col1a1*, *Acta2*, and *Tgfb1* in the liver (n = 4–9). Zero weeks (0W) in the figure indicates Chow-diet-fed control mice. Values are presented as mean ± S.E.M. * *p* < 0.05, ** *p* < 0.01, *** *p* < 0.001, and **** *p* < 0.0001 vs. 0-week group. Statistical significance was calculated by a one-way ANOVA with Dunnett’s post hoc test.

**Figure 4 nutrients-16-04151-f004:**
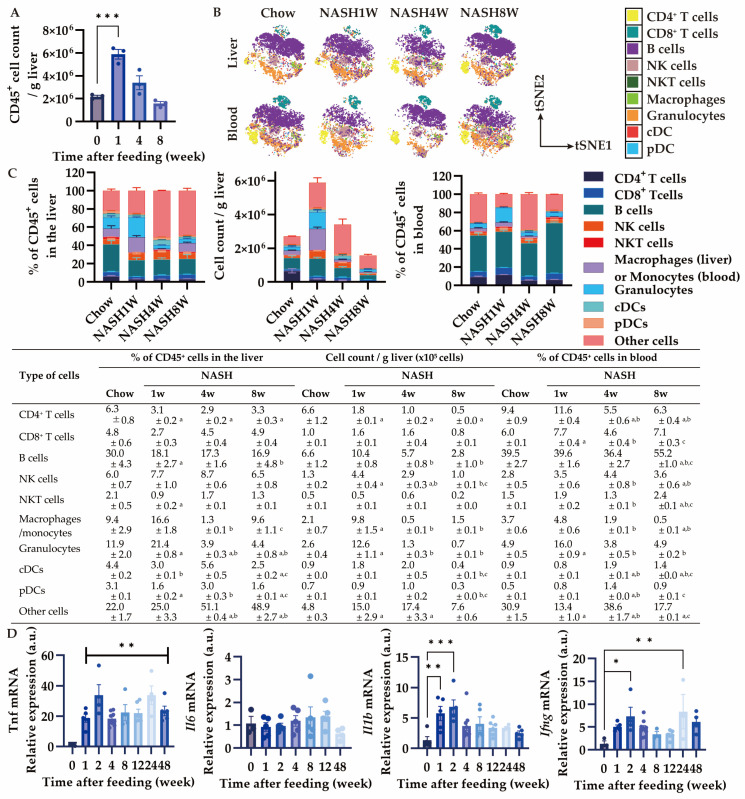
Effects of MASH diet feeding on inflammation in the early stage. (**A**) Total number of CD45^+^ cells in the liver (n = 3); (**B**) representative images of profiling CD45^+^ cells in the liver and blood by CyTOF visualized by tSNE; (**C**) Percentage and the number of CD45^+^ cells in the liver and percentage of the CD45^+^ cells in the blood by CyTOF (n = 3); (**D**) the mRNA expression of *Tnf*, *Il6*, *Il1b*, and *Ifng* in the liver (n = 3–9). Zero weeks (0W) in the figure indicates Chow-diet-fed control mice. Values are presented as mean ± S.E.M. * *p* < 0.05, ** *p* < 0.01, and *** *p* < 0.001 vs. 0-week group in (**A**,**C**); ^a^ *p* < 0.05 vs. Chow group, ^b^ *p* < 0.05 vs. NASH1W, and ^c^ *p* < 0.05 vs. NASH4W group in (**B**). Statistical significance was calculated by a one-way ANOVA with (**A**,**C**) Dunnett’s post hoc test; (**B**) Tukey’s post hoc test.

**Figure 5 nutrients-16-04151-f005:**
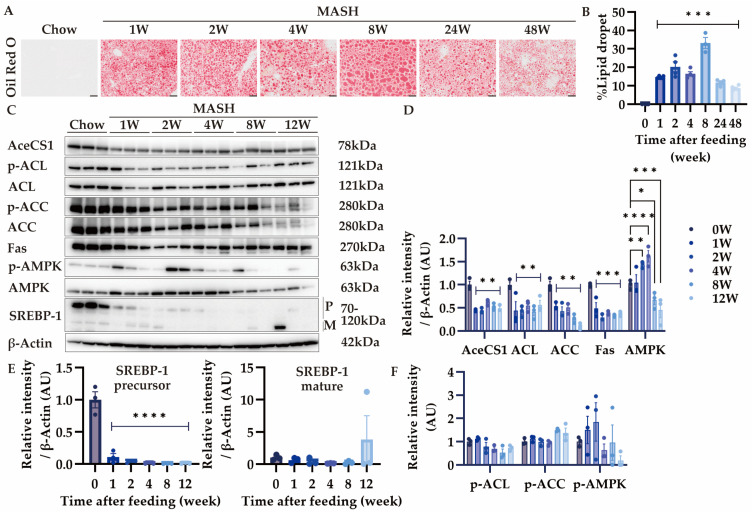
Effects of MASH diet feeding on lipid accumulation and de novo lipogenesis. (**A**,**B**) Oil Red O staining in the liver (n = 3–6); (**C**–**F**) the protein expression of AceCS1, ACL, p-ACL, ACC, p-ACC, Fas, AMPK, p-AMPK, and SREBP-1 in the liver (n = 3). Zero weeks (0W) in the figure indicates Chow-diet-fed control mice. Values are presented as mean ± S.E.M. * *p* < 0.05, ** *p* < 0.01, *** *p* < 0.001, and **** *p* < 0.0001 vs. 0-week group. Statistical significance was calculated by a one-way ANOVA with Dunnett’s post hoc test.

**Figure 6 nutrients-16-04151-f006:**
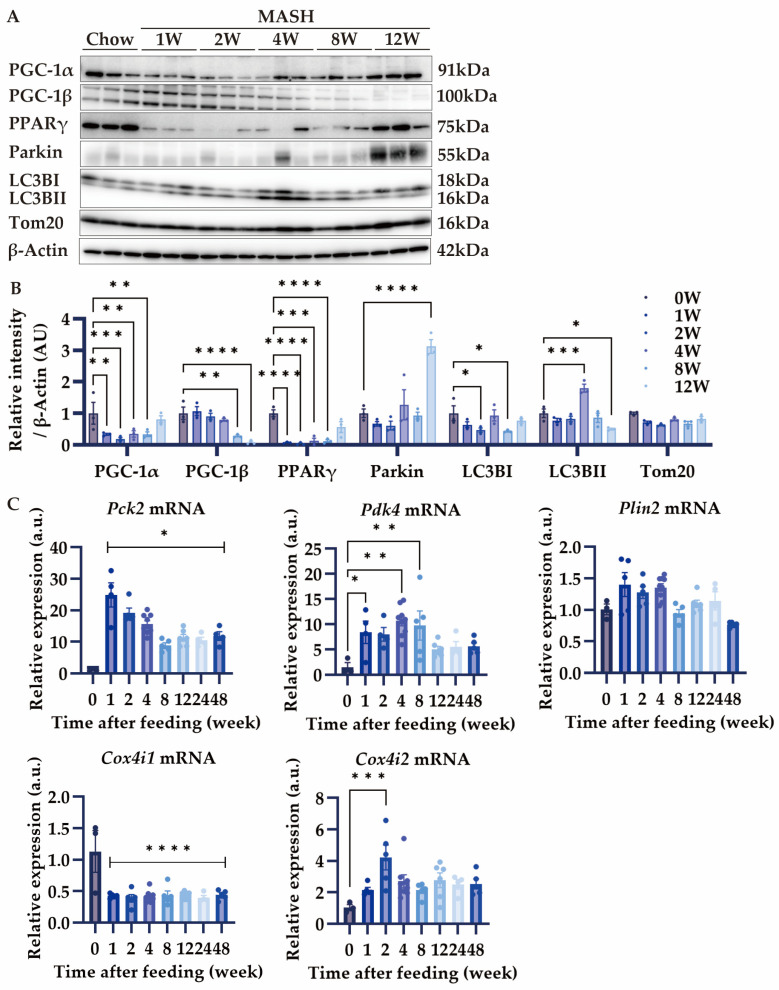
Effects of MASH diet on mitochondrial homeostasis. (**A**,**B**) The protein expression of PGC-1α, PGC-1β, PPARγ, Parkin, LC3B, and Tom20 in the liver (n = 3); (**C**) the mRNA expression of Pck2, Pdk4, Plin2, Cox4i1, and Cox4i2 in the liver (n = 3–9). Zero weeks (0W) in the figure indicates Chow-diet-fed control mice. Values are presented as mean ± S.E.M. * *p* < 0.05, ** *p* < 0.01, *** *p* < 0.001, and **** *p* < 0.0001 vs. 0-week group. Statistical significance was calculated by a one-way ANOVA with Dunnett’s post hoc test.

**Table 1 nutrients-16-04151-t001:** Ingredients of the MASH diet.

Nutrition Facts	MASH Diet: OYC-NASH1
Moisture	9.0%
Crude protein	17.2%
Crude fat	28.2%
Coarse ash content	3.0%
Coarse fiber	4.7%
Nitrogen free extract	38.0%
Calories	474.2 kcal/100 g of the diet
Saturated fatty acid	34.4% of total fatty acid
Monounsaturated fatty acid	29.3% of total fatty acid
Polyunsaturated fatty acid	36.2% of total fatty acid
Methionine	0.11%
Choline	0%

**Table 2 nutrients-16-04151-t002:** List of probes for RT-qPCR.

Probe Name	Probe Sequence (5′ to 3′)
*Col1a1*-forward	GACGCATGGCCAAGAAGACA
*Col1a1*-reverse	ATTGCACGTCATCGCACACA
*Acta2*-forward	CTTCGCTGGTGATGATGCTC
*Acta2*-reverse	GATGATGCCGTGTTCTATCG
*Tgfb1*-forward	TATTTGGAGCCTGGACACAC
*Tgfb1*-reverse	GTAGTAGACGATGGGCAGTGG
*Tnf*-forward	AGCACAGAAAGCATGATCCG
*Tnf*-reverse	GGAGGCCATTTGGGAACTTC
*Il6*-forward	ACAAAGCCAGAGTCCTTCAGAG
*Il6*-reverse	TTGGTCCTTAGCCACTCCTTC
*Il1b*-forward	CCCTGCAGCTGGAGAGTGTGGA
*Il1b*-reverse	TGTGCTCTGCTTGTGAGGTGCTG
*Ifng*-forward	AGACAATCAGGCCATCAGCA
*Ifng*-reverse	TGGACCTGTGGGTTGTTGAC
*Pck2*-forward	ATGGCTGCTATGTACCTCCC
*Pck2*-reverse	GCGCCACAAAGTCTCGAAC
*Pdk4*-forward	AGGGAGGTCGAGCTGTTCTC
*Pdk4*-reverse	GGAGTGTTCACTAAGCGGTCA
*Plin2*-forward	GACCTTGTGTCCTCCGCTTAT
*Plin2*-reverse	CAACCGCAATTTGTGGCTC
*Cox4i1*-forward	GTACCGCATCCAGTTTAACGA
*Cox4i1*-reverse	TGGGGCCATACACATAGCTCT
*Cox4i2*-forward	CTGCCCGGAGTCTGGTAATG
*Cox4i2*-reverse	CAGTCAACGTAGGGGGTCATC
*Hprt*-forward	CGCAGTCCCAGCGTCGTGATT
*Hprt*-reverse	CTTGAGCACACAGAGGGCCACAA

**Table 3 nutrients-16-04151-t003:** Effects of MASH diet on serum biochemical indices.

	Time After Feeding							
Variables	0-Week	1 Week	2 Weeks	4 Weeks	8 Weeks	12 Weeks	24 Weeks	48 Weeks
TP (g/dL)	5.0 ± 0.2	4.6 ± 0.1	4.7 ± 0.1	5.0 ± 0.1	4.6 ± 0.1	4.9 ± 0.1	5.0 ± 0.1	5.2 ± 0.2
Alb (g/dL)	3.2 ± 0.1	3.2 ± 0.1	3.2 ± 0.1	3.3 ± 0.0	3.0 ± 0.1	3.1 ± 0.0	3.1 ± 0.1	3.2 ± 0.1
A/G	1.9 ± 0.0	2.2 ± 0.1	2.2 ± 0.1	2.0 ± 0.1	1.9 ± 0.1	1.7 ± 0.1	1.8 ± 0.2	1.6 ± 0.1
Fe (μg/dL)	180.3 ± 23.1	199.0 ± 7.9	178.3 ± 7.3	182.3 ± 4.8	160.3 ± 9.4	176.7 ± 2.2	166.7 ± 23.5	159.7 ± 7.0
Total Cho (mg/dL)	89.0 ± 2.9	59.0 ± 3.2 ^a^	57.3 ± 3.8 ^a^	65.0 ± 1.2	66.7 ± 5.5	77.0 ± 10.1	91.7 ± 1.7	118.7 ± 11.3 ^a^
Free Cho (mg/dL)	14.7 ± 2.3	19.0 ± 1.2	20.3 ± 1.5	22.7 ± 1.8	20.3 ± 2.3	21.3 ± 2.3	27.3 ± 1.8 ^b^	31.7 ± 2.4 ^d^
Esterified Cho (mg/dL)	75.0 ± 1.5	40.0 ± 3.8 ^c^	37.0 ± 2.5 ^c^	42.3 ± 1.8 ^b^	46.3 ± 3.4 ^b^	55.7 ± 7.8	64.3 ± 2.2	87.0 ± 9.0
E/T (%)	82.3 ± 0.9	67.7 ± 3.3 ^c^	64.3 ± 0.9 ^d^	65.0 ± 2.5 ^d^	69.7 ± 1.2 ^c^	72.0 ± 0.6 ^b^	70.3 ± 1.8 ^b^	73.3 ± 0.9 ^a^
TG (mg/dL)	77.3 ± 2.2	25.3 ± 0.9 ^a^	35.7 ± 6.9	77.3 ± 27.0	56.0 ± 10.5	35.0 ± 6.8	52.7 ± 2.2	30.3 ± 3.8 ^a^
NEFA (μEq/L)	816.3 ± 23.9	805.0 ± 38.2	1100.3 ± 170.5	852.3 ± 148.9	1108.3 ± 61.4	653.7 ± 65.3	945.3 ± 203.1	733.7 ± 79.5
LDL Cho (mg/dL)	6.3 ± 0.3	1.0 ± 0.0 ^b^	1.0 ± 0.0 ^b^	1.0 ± 0.0 ^b^	1.3 ± 0.3 ^b^	1.7 ± 0.3 ^b^	4.7 ± 0.9	7.0 ± 2.1
HDL Cho (mg/dL)	61.0 ± 1.7	33.7 ± 4.1 ^c^	32.0 ± 2.5 ^c^	34.7 ± 1.2 ^c^	35.0 ± 1.2 ^c^	43.0 ± 5.5 ^a^	38.7 ± 2.8 ^b^	63.0 ± 6.2
TKB (μmol/L)	117.0 ± 6.9	639.3 ± 273.9 ^a^	229.7 ± 40.9	236.0 ± 47.5	370.3 ± 136.5	318.7 ± 77.5	2098.3 ± 107.3 ^d^	232.3 ± 45.7

Zero weeks (0-week) in the table indicates Chow-diet-fed control mice. Values are presented as mean ± S.E.M. (n = 3); ^a^ *p* < 0.05, ^b^ *p* < 0.01, ^c^ *p* < 0.001, and ^d^ *p* < 0.0001 vs. 0-week group. Statistical significance was calculated by a one-way ANOVA with Dunnett’s post hoc test. TP: total protein; Alb: albumin; A/G: albumin/globulin ratio; Cho: cholesterol; E/T: esterified cholesterol/total cholesterol; TG: Triglyceride; NEFA: non-esterified fatty acid; LDL: low-density lipoprotein; HDL: high-density lipoprotein; TKB: total ketone body.

## Data Availability

Data can be obtained from the corresponding author upon reasonable request.

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
