# Peer review of "Novel Choline-Deficient and 0.1%-Methionine-Added High-Fat Diet Induces Burned-Out Metabolic-Dysfunction-Associated Steatohepatitis with Inflammation by Rapid Immune Cell Infiltration on Male Mice"

_nutrients, 2024, doi:10.3390/nu16234151_

Round 1
Reviewer 1 Report
Comments and Suggestions for Authors
This study was to investigate the effect on an innovative diet on the induction of metabolic dysfunction associated steatohepatitis (MASH) that could recapitulates the key features of human MASH, including the progression of steatosis, inflammation, and fibrosis.
This study used mice with a 0.01% of difference in methionine components in the diet when compared to previous used diet. This modification of diet aimed to induce a more rapid and robust progression of MASH, without causing severe weight loss, which is often associated with traditional CDAHFD models.
5 weeks old male mice were divided into two groups that supplemented with standard chow diet or modified choline-deficient, 0.11% methionine-added high-fat diet (MASH diet). Then At 0, 1, 2, 4, 8, 12, 24, and 48 weeks after the feeding, the animals were weighed and euthanized. Blood was collected for analysis (CyTOF analysis & Biochemical analysis. Liver Tissue was collected for histological ((H&E & sirius red), gene expression analysis (Real time PCR), and protein analysis (Western blot and CyTOF analysis.
The research team showed a significant increase in early steatosis and fibrosis than the traditional CDAHFD models. CD45 positive cells only increased in the first week. The author also showed changed immune cell subpopulation.
With the modified diet it showed a high lipid accumulation with a suppression of DNL in liver at week of 12. From the serum components analysis, with the decrease of serum cholesterol, LDL at 12 weeks of feeding they showed that the export of cholesterol from the liver is low. However, they showed SREBP-1 expression also low that means the liver is producing less cholesterol and it developed burned out MASH.
While this study provided comprehensive analysis from morphological changed to cellular and molecular modification related to metabolic disfunction and immunological changes, but more efforts are needed for making the manuscript easy to follow and highlight what are the differences from the previous diet.
Major points
· Figure 3B and 3C differences cannot be read from the figure. Author should consider using a table (with indication of significance) to match the text part of the results (line 254-259)
· The study was done only on male not on female. Female mice model could be included to assess sex-specific differences, or the study title should clearly mention that the research was done only on males.
· A clear and concise experimental design diagram should be included in the Methods section.
· On page 3, the author mentioned that 5 weeks old mice were taken, however on page 7 they mentioned 6 weeks old mice were taken for this study. So, the age of mice should be consistent throughout the manuscript.
· In the methods section, the study describes two groups of mice, one control group and other one fed the modified diet. However, while control group data are present on the histological images as well as in the Western blot bands (Figures1, 2, 4 & 5), they are missing in the corresponding graphical representations. This inconsistency should be addressed to ensure accurate comparison between two groups
· The study does not specify the feeding timeline for the control group or indicate the specific week at which the control animals were sacrificed. This information is important for ensuring accurate comparisons with the experimental group.
· In Figures 4 and 5, the β-actin bands look identical. Additionally, the raw data of the corresponding images for is missing, this is not acceptable unless the author could provide β-actin matching each blot with original image
· On page 2 under introduction, the author mentions that they develop a novel MASH diet with some changes from the ingredients of the previously reported CDAHFD diet, but it is not specifically mentioned what changes so a side-by-side comparison between the two diet components would be easier to understand
- In Figure 3, the author presents a graphical representation of CyTOF data, however, a CyTOF cell population image from the original instruments are needed
- On the discussion part to explain the steatosis, inflammation and fibrosis result a comparison between the conventional CDAHFD diet and this modified MASH diet would be easier to understand
Minor points
· Some correction on the spacing, font size (not consistent) and figure formatting
· On the material and method part the author is mentioning the modified CHFDA diet as the MASH diet, however they should mention it as the MASH diet in a bracket on the abstract too.
· The flow of the writing seems a little bit off, for example, in line 208-212, there are a paragraph talking about previous research, which should be in the discussion and should point out what are the differences between MASH diet and CHFDA diet. Another part is line 251-254, the author was describing their results, and jumped to a previous research and the sentence are very confusing, with no connection to the figure 3B results. I have no idea why the author should put this statement here. I highly recommend the research team to do a comprehensive edit on writing. Especially the results and discussion part.
Comments on the Quality of English Language
The English is acceptable, but the explaination of results and discussion needs to be improved. the logic and flow from the results and discussion are not going smoothly.
Author Response
We express our gratitude to all the reviewers for conducting an in-depth review of the manuscript and for providing constructive and positive feedback. Their thoughtful comments and suggestions have played a crucial role in refining and strengthening our work. The revisions made during this process are highlighted in blue characters for clarity:
Responses from Reviewer-1
Major points
Figure 3B and 3C differences cannot be read from the figure. Author should consider using a table (with indication of significance) to match the text part of the results (line 254-259).
According to your suggestion, we listed and inserted the table (with the indication of significance) for each immune cell population’s proportion and actual number based on the CyTOF analysis as Figure 4C.
The study was done only on male not on female. Female mice model could be included to assess sex-specific differences, or the study title should clearly mention that the research was done only on males.
We mentioned in the title that the research was done only on male mice.
A clear and concise experimental design diagram should be included in the Methods section.
We inserted an experimental design diagram for all the experiments in the Method section as Figure 1.
On page 3, the author mentioned that 5 weeks old mice were taken, however on page 7 they mentioned 6 weeks old mice were taken for this study. So, the age of mice should be consistent throughout the manuscript.
We purchased the 5-week-old mice from a vendor and let them acclimate for 1 week. Then, we started using them for experiments from 6 weeks old. We described this in section 2.1 (lines 92-94).
In the methods section, the study describes two groups of mice, one control group and other one fed the modified diet. However, while control group data are present on the histological images as well as in the Western blot bands (Figures 1, 2, 4 & 5), they are missing in the corresponding graphical representations. This inconsistency should be addressed to ensure accurate comparison between two groups.
We corrected the Chow description in all the Figures and Table 3 to 0W. Additionally, we mentioned in each figure and table legend that 0W indicated Chow diet-fed mice.
The study does not specify the feeding timeline for the control group or indicate the specific week at which the control animals were sacrificed. This information is important for ensuring accurate comparisons with the experimental group.
Actually, we sacrificed and collected the samples from both control groups (Chow diet) and the MASH diet-fed group at each time point. However, when we compared the 0W and 48W of the Chow diet-fed group, we identified no significant changes in serum biochemical analysis between them, while body weight showed the difference. This data indicates the less phenotypic difference of Chow diet-fed mice through the entire period. Furthermore, this result suggests that the data of each time point Chow diet gives no meaningful information to the readers, and we decided to show only 0W Chow control data to simplify the data visualization. We show the result of serum biochemical analysis for Chow 0W and 48W below, but we will not show it in our manuscript.
Table. Serum biochemical indices between Chow 0W and 48W.
Chow diet |
0w |
48w |
p-value |
TP (g/dL) |
5.0 ± 0.2 |
5.5 ± 0.2 |
0.15 |
ALB (g/dL) |
3.2 ± 0.1 |
3.4 ± 0.2 |
0.34 |
A/G |
1.9 ± 0.0 |
1.6 ± 0.1 |
0.12 |
Fe (μg/dL) |
180.3 ± 23.1 |
154.3 ± 11.6 |
0.39 |
AST (IU/L) |
60.0 ± 15.6 |
45.3 ± 4.5 |
0.45 |
ALT (IU/L) |
20.7 ± 1.2 |
25.3 ± 3.4 |
0.30 |
Total-Cho (mg/dL) |
89.0 ± 2.9 |
90.0 ± 9.2 |
0.93 |
Free-Cho (mg/dL) |
14.7 ± 2.3 |
13.7 ± 0.9 |
0.72 |
Esterified-Cho (mg/dL) |
75.0 ± 1.5 |
76.3 ± 8.4 |
0.89 |
E/T (%) |
82.3 ± 0.9 |
84.7 ± 0.7 |
0.11 |
TG (mg/dL) |
77.3 ± 2.2 |
97.3 ± 14.4 |
0.30 |
NEFA (μEq/L) |
816.3 ± 23.9 |
866.7 ± 145.0 |
0.76 |
LDL-Cho (mg/dL) |
6.3 ± 0.3 |
5.3 ± 0.9 |
0.38 |
HDL-Cho (mg/dL) |
61.0 ± 1.7 |
62.7 ± 5.8 |
0.81 |
T-KB (μmol/L) |
117.0 ± 6.9 |
102.7± 9.0 |
0.28 |
Values are presented as mean ± S.E.M. (n=3); Statistical significance was calculated by Student’s t-test. TP: total protein; Alb: albumin; A/G: albumin/globulin ratio; AST: aspartate aminotransferase; ALT: alanine aminotransferase; Cho: cholesterol; E/T: esterified cholesterol/total cholesterol; TG: Triglyceride; NEFA: non-esterified fatty acid; LDL: low density lipoprotein; HDL: high density lipoprotein; TKB: total ketone body
In Figures 4 and 5, the β-actin bands look identical. Additionally, the raw data of the corresponding images for is missing, this is not acceptable unless the author could provide β-actin matching each blot with original image.
We apologize for our fault and appreciate for pointing out this issue. We changed the β-actin control bands of Figure 6 (Figure 5 of the previous version) to the corrected one.
On page 2 under introduction, the author mentions that they develop a novel MASH diet with some changes from the ingredients of the previously reported CDAHFD diet, but it is not specifically mentioned what changes so a side-by-side comparison between the two diet components would be easier to understand.
CDAHFD diet is commercially provided by many companies, and all the companies do not disclose the ingredients of their diet in detail. Because of this, it is difficult to show a detailed comparison of the CDAHFD diet and our MASH diet side by side. However, to follow your suggestion, we described our modification in the Introduction section (lines 81-83).
In Figure 3, the author presents a graphical representation of CyTOF data, however, a CyTOF cell population image from the original instruments are needed.
We added the tSNE plots for CyTOF analysis in Figure 4B (Figure 3 of the previous version).
On the discussion part to explain the steatosis, inflammation and fibrosis result a comparison between the conventional CDAHFD diet and this modified MASH diet would be easier to understand.
We corrected our discussion part to follow your suggestion that the readers can easily understand the phenotypic difference between the CDAHFD diet and the MASH diet.
Minor points
- Some correction on the spacing, font size (not consistent) and figure formatting.
We corrected the spacing, font size, and figure formatting for all the figures and tables.
- On the material and method part the author is mentioning the modified CHFDA diet as the MASH diet, however they should mention it as the MASH diet in a bracket on the abstract too.
We described the MASH diet in the abstract according to your suggestion (line 23).
- The flow of the writing seems a little bit off, for example, in line 208-212, there are a paragraph talking about previous research, which should be in the discussion and should point out what are the differences between MASH diet and CHFDA diet. Another part is line 251-254, the author was describing their results, and jumped to a previous research and the sentence are very confusing, with no connection to the figure 3B results. I have no idea why the author should put this statement here. I highly recommend the research team to do a comprehensive edit on writing. Especially the results and discussion part.
We carefully edited the sentences you pointed out to avoid misleading readers. Some of the sentences that include the previous reports, we deleted from the result part and described in the discussion part to improve the flow.

Reviewer 2 Report
Comments and Suggestions for Authors
Introduction
The article makes a significant contribution to the development of preclinical models for MASH. The introduction of a novel dietary model and the use of CyTOF for immunological analysis are strong aspects of the study. However, the text requires some stylistic and structural revisions to enhance readability. I recommend emphasizing the originality of the approach and the modifications introduced in the diet more clearly. Shortening sentences and avoiding repetitions would improve the text's flow. Precise language is needed to eliminate unnecessary ambiguity. For example: “Metabolic dysfunction associated steatotic liver disease (MASLD) is the current term for liver disease associated with metabolic dysfunction and steatosis [1]. Metabolic dysfunction associated steatohepatitis (MASH) is also the latest term that represents steatohepatitis that possesses fibrosis like non-alcoholic steatohepatitis (NASH) which was previously termed [1].”
Materials and Methods
Some sections repeat procedures or provide excessive details that could be streamlined. For example:“Blood was collected by cardiac puncture into a serum collection tube (#365978, BD Bioscience) and allowed to clot at room temperature for 30 min. After centrifugation for 90 s at 6000 × g at 4°C, serum was collected and stored at -80°C.” could be revised to: “Blood was collected by cardiac puncture, clotted at room temperature for 30 minutes, centrifuged (6000 × g, 4°C, 90 s), and stored at -80°C.” Some elements, such as reagent descriptions, lack consistency. In certain cases, catalog numbers are provided, while in others, they are omitted. This should be standardized, e.g., "ACK Lysing Buffer (#A1049201, Fisher Scientific)" or "ACK Lysing Buffer, Fisher Scientific, #A1049201." Certain information (e.g., "Table 1" with nutritional values of the diet) is embedded within procedural descriptions, which can hinder readability. I suggest placing the table in a separate subsection titled “Nutritional Composition of the MASH Diet” (2.1.1). Descriptions of protocols (e.g., MASH diet, cell isolation, sample preparation for Western blot) could benefit from diagrams or flowcharts summarizing the process. This would be especially helpful for more complex procedures.
Results
The description of Figure 1 references various panels (A–H) but does not always clearly indicate which conclusions correspond to specific plots. Providing more explicit references to individual panels in the text would improve clarity. Data from the plots (e.g., NAS, ALT, AST) are described accurately, but readability could be further enhanced. Avoid overly lengthy figure captions—shorter and more focused captions would be more effective.
Discussion
The Discussion section effectively highlights the importance of the findings and their place in the literature, but it could be expanded to include more practical implications for both basic research and potential clinical applications.
Conclusions
The statement that the MASH model induces fibrosis faster than the CDAHFD diet represents a significant advancement in preclinical models. However, it would be valuable to elaborate
Author Response
We express our gratitude to all the reviewers for conducting an in-depth review of the manuscript and for providing constructive and positive feedback. Their thoughtful comments and suggestions have played a crucial role in refining and strengthening our work. The revisions made during this process are highlighted in blue characters for clarity:
Responses from Reviewer-2
Introduction
The article makes a significant contribution to the development of preclinical models for MASH. The introduction of a novel dietary model and the use of CyTOF for immunological analysis are strong aspects of the study. However, the text requires some stylistic and structural revisions to enhance readability. I recommend emphasizing the originality of the approach and the modifications introduced in the diet more clearly. Shortening sentences and avoiding repetitions would improve the text's flow. Precise language is needed to eliminate unnecessary ambiguity. For example: “Metabolic dysfunction associated steatotic liver disease (MASLD) is the current term for liver disease associated with metabolic dysfunction and steatosis [1]. Metabolic dysfunction associated steatohepatitis (MASH) is also the latest term that represents steatohepatitis that possesses fibrosis like non-alcoholic steatohepatitis (NASH) which was previously termed [1].”
We carefully reviewed our manuscript and edited the introduction section based on your suggestions.
Materials and Methods
Some sections repeat procedures or provide excessive details that could be streamlined. For example:“Blood was collected by cardiac puncture into a serum collection tube (#365978, BD Bioscience)and allowed to clot at room temperature for 30 min. After centrifugation for 90 s at 6000 × g at 4°C, serum was collected and stored at -80°C.” could be revised to: “Blood was collected by cardiac puncture, clotted at room temperature for 30 minutes, centrifuged (6000 × g, 4°C, 90 s), and stored at -80°C.” Some elements, such as reagent descriptions, lack consistency. In certain cases, catalog numbers are provided, while in others, they are omitted. This should be standardized, e.g., "ACK Lysing Buffer (#A1049201, Fisher Scientific)" or "ACK Lysing Buffer, Fisher Scientific, #A1049201." Certain information (e.g., "Table 1" with nutritional values of the diet) is embedded within procedural descriptions, which can hinder readability. I suggest placing the table in a separate subsection titled “Nutritional Composition of the MASH Diet” (2.1.1). Descriptions of protocols (e.g., MASH diet, cell isolation, sample preparation for Western blot) could benefit from diagrams or flowcharts summarizing the process.
We carefully edited our Materials and Methods section to simplify the explanation and standardize the catalog number description. We also separated Table 1 from “2.1. Animals and experimental protocol for the MASH diet model” and made a subsection titled “2.1.1 Nutritional Composition of the MASH Diet” according to your suggestion. We further added the diagrams for each experiment in Figure 1.
Results
The description of Figure 1 references various panels (A–H) but does not always clearly indicate which conclusions correspond to specific plots. Providing more explicit references to individual panels in the text would improve clarity. Data from the plots (e.g., NAS, ALT, AST) are described accurately, but readability could be further enhanced. Avoid overly lengthy figure captions—shorter and more focused captions would be more effective.
We carefully edited our description to make it easy for readers to understand all the figures.
Discussion
The Discussion section effectively highlights the importance of the findings and their place in the literature, but it could be expanded to include more practical implications for both basic research and potential clinical applications.
We edited our discussion to imply the benefit of our MASH diet both for basic research and the pre-clinical model for drug discovery. We also pointed out the similarity between the pathophysiology of our MASH diet and that of human MASH patients to suggest potential clinical implications.
Conclusions
The statement that the MASH model induces fibrosis faster than the CDAHFD diet represents a significant advancement in preclinical models. However, it would be valuable to elaborate.
We edited the sentence to emphasize the contribution of myeloid cell infiltration and inflammation into the liver associated with the MASH diet feeding.
